# PD-L1/PD-1 Expression in the Treatment of Oral Squamous Cell Carcinoma and Oral Potentially Malignant Disorders: An Overview of Reviews

**DOI:** 10.3390/jpm15040126

**Published:** 2025-03-25

**Authors:** Huda Moutaz Asmael Al-Azzawi, Syed Ameer Hamza, Rita Paolini, Mathew Lim, Romeo Patini, Antonio Celentano

**Affiliations:** 1Melbourne Dental School, The University of Melbourne, 720 Swanston Street, Carlton, VIC 3053, Australia; hasmael@student.unimelb.edu.au (H.M.A.A.-A.); h.hamza@student.unimelb.edu.au (S.A.H.); rita.paolini@unimelb.edu.au (R.P.); mathew.lim@unimelb.edu.au (M.L.); 2Head and Neck Department, “Fondazione Policlinico Universitario A. Gemelli—IRCCS” School of Dentistry, Catholic University of Sacred Heart—Rome Largo A. Gemelli, 8, 00168 Rome, Italy; romeo.patini@unicatt.it

**Keywords:** programmed death ligand 1, oral squamous cell carcinomas, oral potentially malignant disorders (OPMDs), biomarker, oral cancer, programmed death receptor 1

## Abstract

**Objective**: In this overview, we present compelling evidence from multiple systematic reviews and meta-analyses (SRMAs) and examine the prognostic role of the PD-L1/PD-1 axis, as well as the potential of personalized treatment strategies targeting this axis, in patients with oral squamous cell carcinoma (OSCC) and oral potentially malignant disorders (OPMDs). **Methods**: Six databases were searched to retrieve systematic review and meta-analysis studies. The population of interest was patients with OSCC and OPMDs in whom the expression of PD-L1 and PD-1 had been investigated. At least one of the following outcomes was reported, along with at least one clinicopathological feature: overall survival, disease-free survival, or disease-specific survival. All studies were assessed for risk of bias using the AMSTAR 2 tool. **Results**: A total of 195 studies were found through the initial search, and after duplicate removal, 97 studies were screened by title and abstract. Finally, five systematic reviews and meta-analysis studies fit our inclusion criteria and were included in this review. **Conclusions**: Based on two published systematic reviews, our study revealed a lack of evidence for the prognostic value of PD-L1 in improving overall survival in oral cancer patients. However, it showed a correlation with specific clinicopathological features such as sex, lymph node metastasis, and HPV status.

## 1. Introduction

Oral cancer stands out as the most common cancer in the head and neck region, where oral squamous cell carcinoma accounts for over 90% of oral cancer [1]. Lip and Oral cavity cancers accounted for 389,485 new cases and 188,230 deaths in 2022 [2]. For the majority of oral cancer cases, surgery is the primary treatment. Adjuvant treatment, including radiotherapy and chemotherapy, is employed in cases of distant or locoregional metastasis [3]. Immunotherapy has emerged as a new therapeutic option with promising results against oral cancer. Patient selection for this therapy is the primary factor for the determination of its effectiveness [4]. Current immunotherapy includes multiple options such as immune checkpoint inhibitors (CPIs), costimulatory agonists, adoptive T cell transfer (ACT), oncolytic virus therapy, antigenic vaccines, and epidermal growth factor receptor (EGFR)-targeted therapy [5].

In 2016, the FDA approved immunotherapy as an adjuvant treatment for metastatic squamous cell carcinoma of the head and neck, following successful clinical trials of pembrolizumab and nivolumab. All of these medications are anti-programmed death receptor 1 (PD-1) antibodies [6,7,8]. Program death ligand 1 (PD-L1) is a membrane bound protein recognized as an inhibitory immune check point in the immune response. It is expressed on number of cells, including tumour cells and inflammatory cells (activated B and T cells). When it interacts with its receptors, PD-1 reduces proliferation of PD-1 positive cells When PD-L1 interacts with its receptors PD-1, it reduces the proliferation of PD-1 positive cells, as illustrated in Figure 1. This interaction is behind the mechanism of immune evasion by modulating T-cell activity, inducing apoptosis of antigen-specific T cells and inhibiting apoptosis of regulatory T cells [9]. Additionally, the tumour itself can induce expression of PD-L1 on other cells, resulting in reduction of T-cell activity in tumour microenvironment [10]. PD-L1 is favoured over PD-1 as a predictive biomarker for immunotherapy response because it is more consistently assessed with immunohistochemistry (IHC). On the other hand, while PD-1 is assessed only in inflammatory cells, PD-L1 can be detected on both tumour and inflammatory cells [11]. Tumour PD-L1 expression is associated with prognosis and response to immunotherapy in many malignancies such as gastric cancer [12], colorectal cancer [13], renal cell cancer [14], and non-small cell lung cancer [15].

The PD-L1/PD-1 axis plays an important role in T-cell exhaustion, which occurs when T cells encounter cancer cells continuously, leading to weakened T cells and an increase in the expression of inhibitory checkpoint proteins on their surface, such as PD-1 molecules. This results in less effective T cells in the tumour microenvironment. The PD-L1/PD-1 axis is a crucial component in T-cell exhaustion. Blocking this axis with checkpoint inhibitors can boost the immune system’s response against cancer [16].

In oral cancer, the PD-L1 immune checkpoint acts as a tumour biomarker that can help in selecting patients for immunotherapy; monitoring patient response to immunotherapy and in assessing tumour prognosis and is correlated with clinicopathological features [17]. The administration of checkpoint inhibitor therapy in head and neck squamous cell carcinoma (HNSCC) requires immunohistochemical identification of the PD-L1 molecule. The utilization of monoclonal antibodies against immune checkpoint molecules like PD-L1 activates the immune response to tumours and consequently improves the outcome and prognosis [18]. Oral potentially malignant disorders (OPMDs) encompass a range of clinically recognized conditions with differing risks of developing into oral squamous cell carcinoma (OSCC). These lesions are typically visible upon clinical examination [19]. Studies indicate that elevated PD-L1 levels may be linked to the progression of OPMDs into malignancy [20,21].

OSCC is a highly aggressive disease with rising incidence. Therefore, enhancing therapeutic approaches for OSCC patients is an urgent necessity. The role of PD-L1 in tumour immunity and its potential indicator as a biomarker for poor prognosis in OSCC is still controversial, due to IHC challenges in the detection of PD-L1 and PD-1. These challenges include a lack of standardization in assessment methods across different studies, such as variations in assay types, antibodies used, and cutoff values. These factors contribute significantly to the controversial results regarding the role of PD-L1 in OSCC [22,23,24], as illustrated in Table 1.

While much is known about the PD-L1/PD-1 axis in various types of cancer, its specific role in OSCC and OPMDs remains unclear, particularly regarding its prognostic value, response to treatment, and clinicopathological features. This highlights the gap in knowledge regarding the clinical utility of PD-L1 in OSCC and OPMDs. In this overview, we analyse the existing scientific literature on the prognostic role of the PD-L1/PD-1 axis by presenting data from systematic reviews and meta-analyses (SRMA). A further objective will be to investigate the efficacy of treatments that interact with this axis in patients with OSCC and OPMD.

## 2. Materials and Methods

A research protocol for this umbrella review was created and followed according to Preferred Reporting Items for Overviews of Reviews (PRIOR) [25]. All SRMAs included were assessed for methodological quality using the AMSTAR 2 tool [26]. Moreover, the mapping, ordering, overview, and recommendation (MOoR) framework was utilized to resolve primary studies overlap.

**Inclusion criteria**: We used the PICOS criteria for determining the eligibility of the studies for inclusion, which includes the following:Populations: Patients with OSCC and OPMDs.

For this study, oral potentially malignant disorders (OPMDs) were defined based on the latest classification by the WHO Collaborating Centre for Oral Cancer, including leukoplakia, erythroplakia, proliferative verrucous leukoplakia, oral lichen planus, oral lichenoid lesions, oral submucous fibrosis, palatal lesions in reverse smokers, lupus erythematosus, epidermolysis bullosa, dyskeratosis congenita, and oral manifestations of chronic graft-versus-host disease [27].

2.Intervention: The expression of PD-L1 and PD-1.3.Comparator: Normal healthy individuals.4.Outcome: The presence of at least one of the following outcomes: overall survival (OS), disease-free survival (DFS), disease specific survival (DSS), progression-free survival (PFS), TMN status, and histological grade.5.Studies: Only SRMAs in English were included, without any restriction on the publication date.


**Exclusion criteria:**
Studies that included head and neck cancer subsets other than oral cancer.Oral cavity tumours other than OSCC.Animal studies and in vitro studies.Studies about oral cancer mentioned as a subset along with other types of body cancer.Studies that assessed markers post treatment, such as post-immune-check-point inhibitors, radiotherapy, magnetic resonance imaging (MRI), etc., as these studies monitored treatment response and have a different aim.Studies about PD-L1 and PD-1 as a subset within other biomarkers



**Search strategy**


A systematic search of the literature for systematic reviews and meta-analysis studies (SRMAs) was conducted on 25 May 2024 using the following six databases: Medline via Ovid, All EBM (Cochrane Library), Embase (Ovid), CINAHL (EBSCO), Web of Science Core Collection, and Scopus. Two Boolean operators (AND, OR) were used in the search for the keywords. The keywords utilized in this study are provided as a Appendix A: Keywords used in our study. Articles were considered eligible if they included meta-analyses conducted with systematic methods.


**Primary and secondary outcomes**


The primary outcomes were OS, DFS, DSS, and/or lymph node metastasis. Secondary outcomes were recurrence, TNM status, human papillomavirus (HPV) status and sex.


**Study selection process and data extraction**


An electronic search for studies across the databases was conducted by two independent reviewers (H.A and S.H). Conflicts between reviewers were solved by a third investigator (A.C). Data from the studies were extracted using a standardized data extraction Microsoft^®^ Excel^®^ form. The Excel file contained details about the study characteristics: first author surname, year, country, number of studies included in the SRMA, study design, number of patients involved, method of OSCC diagnosis, method of PD-L1 assessment, OS, DFS, DSS, PFS, lymph node metastasis, recurrence, TNM status, HPV status, sex, and main findings.

All SRMAs were stratified for subgroup analysis of heterogeneity for each outcome. An I^2^ value below 50% represented low or moderate heterogeneity while a value above 50% represented high heterogeneity.


**Statistical analyses**


Descriptive analysis was conducted by transferring the data to Microsoft^®^ Excel^®^ for Microsoft 365 MSO (Version 2403 Build 16.0.17425.20176). Absolute percentages, Cohen’s kappa coefficient, and inter-rater agreement were calculated using IBM SPSS Statistics for Windows, version 42 (IBM Corp., Armonk, NY, USA). The MOoR framework was used to resolve the overlap in the primary studies involved in the included SRMAs.

## 3. Results

A total of 195 studies were found through initial search and imported into Covidence. After removing 98 duplicates, 97 records were screened by title and abstract. After resolving conflicts between reviewers, 75 studies were excluded. The probability of agreement between reviewers was po = −97.9% with a Cohen’s kappa of 0.941, indicative of perfect agreement. A total of 22 articles were chosen for full text article screening, with 17 Records disqualified from further analysis due to various considerations (Figure 2). The probability of agreement between reviewers was po = 95.45%, with Cohen’s kappa of 0.878, again indicative of almost perfect agreement. Risk of bias for SRMAs is provided in a Appendix A: Risk of bias using AMSTAR 2 for SRMA studies. 

### 3.1. Studies Characteristics

A total of five SRMAs were included for data extraction. The studies were published between 2019 and 2023 from different countries: two studies from Italy, two from China, and one from Spain. All studies assessed PD-L1 expression in tissue of OSCC patients by means of IHC. Following our stringent criteria, we did not identify any SRMAs evaluating the PD-L1/PD-1 axis in OPMDs. Therefore, our analysis will focus on our findings regarding PD-L1/PD-1 expression in OSCC.

The number of studies included in these SRMAs ranged from 11 to 26 studies and the number of patients ranged from 1060 to 3217. The majority of patients in the included studies were of Asian and Caucasian backgrounds.

The outcomes assessed in these studies include the role of PD-L1 in the prognosis and clinicopathological features of OSCC, expressed as OS, DFS, DSS, lymph node metastasis, distal metastasis, HPV infection status, IHC findings, TNM stage, tumour recurrence, cutoff value, and sex.

Based on our subgroup analysis for heterogeneity, using I^2^ values, almost all of the SRMAs had a high rate of heterogeneity between the included studies. We stratified them for subgroup analysis of heterogeneity for each outcome. Where an I^2^ value above 50% represented high heterogeneity, only two of the included studies had an I^2^ below this threshold.

### 3.2. Studies Overlap

We used the structured approach of the MOoR framework to resolve the overlap in the primary studies involved in these SRMAs. The MOoR framework identified nine methods to manage the overlap used across four steps in the conduct of an overview [28]. Following recommendations of the MOoR framework, we extracted all the SRMAs at this stage (Table 2) and managed the overlap later [29]. We used two non-statistical methods to resolve the overlap at this stage. We decided to exclude two studies [17,30] because of the complete overlap in their primary studies with the other included SRMA studies.

### 3.3. Risk of Bias

Risk of bias for the SRMAs was employed using the AMSTAR 2 tool, as demonstrated in Figure 3. All studies (100%) adequately addressed the research question and inclusion criteria, performed study selection and data extraction in duplicate, and utilized appropriate methods to combine meta-analysis results. All studies (100%) failed to explain the selection of study design included in their review and did not report the source of funding for the included primary studies. Only 40% of studies established a clear protocol and methods prior to conducting their review, explained the reasons behind heterogeneity of the included studies, and account for risk of bias (ROB) in individual studies when interpreting/discussing the results. Eighty percent of the studies provided a list of the excluded studies, justified their exclusions, and performed meta-analysis on low risk of bias studies. Moreover, 60% of the studies performed risk of bias assessment for the included studies and assessed ROB effect on the results. At this stage, one study was excluded from further analysis because of high risk of bias [11].

### 3.4. Summary of the Findings

We excluded two studies because of complete overlap [17,30], and one study due to high risk of bias [11]. Finally, we included two high quality studies to present their main findings in this section [22,31].

#### 3.4.1. The Prognostic Value of PD-L1, Including Overall Survival and Disease-Free Survival

The results from two SRMAs that evaluated the role of PD-L1 and OS in OSCC patients showed that there was no significant correlation between PD-L1 and OS [22,31]. However, the level of heterogeneity was high among these studies, as it reached an I^2^ value of 74% (*p* < 0.00001), indicating that the studies’ results are strongly conflicting with each other [22]. Seventeen studies involved 2435 patients with OSCC that were evaluated for PD-L1 expression and OS. The results showed that positive PD-L1 expression was not significantly associated with OS in OSCC patients (HR = 1.00; 95% CI = 0.76–1.30; *p* = 0.284) [31]. Similarly, there was a lack of correlation between PD-L1 and DFS in one SRMA [22].

#### 3.4.2. Lymph Node Metastasis and TNM Stage

High PD-L1 expression was significantly associated with lymph node metastasis in one SRMA, which included 17 studies with 2291 OSCC patients. It showed that positive PD-L1 expression was significantly related to lymph node metastasis status (RR = 0.83; 95% CI = 0.76–0.91; *p* < 0.001), with no apparent heterogeneity between the included studies [31]. While it was close to significant in the other study, the heterogeneity amongst the studies included was moderate (I^2^ = 47%; *p* = 0.02) [22].

To evaluate the relationship between PD-L1 expression and TNM stage, 13 studies involving 1979 patients were analysed [31]. Among the 632 patients with TNM stage I–II, 284 exhibited positive PD-L1 expression, while among the 1347 cases of TNM stage III–IV, 735 cases were positive for PD-L1. The positive rate in patients with TNM stage I–II (44.9%) was lower than in those with TNM stage III–IV (54.6%). These findings indicate a significant association between positive PD-L1 expression and TNM stage (RR = 0.81; 95% CI = 0.73–0.89; *p* < 0.001) [31].

#### 3.4.3. Sex and PD-L1 Expression

In these two SRMAs, there was a significant correlation between high PD-L1 expression and sex. Significant high levels of PD-L1 were associated with female sex (28, 29). The level of heterogeneity was high in one study [31].

#### 3.4.4. HPV Status and PD-L1 Expression

Only two SRMAs, one of which was excluded from this review due to overlap [17,31], investigated the relationship between PD-L1 and HPV status in OSCC cases. In this SRMA, the authors found a significant relationship and a high level of expression of PD-L1 and OSCC in HPV-positive cases. The level of heterogeneity was high, being I^2^ = 69.3% [31].

#### 3.4.5. Expression of PD-L1 and Tumour Recurrence

In one SRMA comparing the high PD-L1 expression in 536 patients from six studies, the difference was not significant (*p* = 0.205) and there was no apparent heterogeneity among the included studies [31].

#### 3.4.6. Histological Grade and PD-L1 Expression

High PD-L1 expression was significantly associated with advanced histological grade (poorly/moderately differentiation) [31]. PD-L1 expression was evaluated in 14 studies among OSCC patients with varying histological differentiation. Among 1357 patients with poor/moderate differentiation, 704 were positive for PD-L1 (51.9%). In comparison, among 503 patients with good differentiation, 248 were positive for PD-L1 (49.3%). The findings indicated that the positive PD-L1 expression was significantly different between the two groups (RR = 1.15; 95% CI = 1.02–1.30; *p* = 0.020), and no apparent heterogeneity was found between the studies [31].

#### 3.4.7. The Cutoff Value of PD-L1 Expression

Cutoff values were varied among primary studies including >1%, >2%, >5%, >10%, >20%, and >25%, where the most commonly used were >2% and >5% [22,31]. The selection of a cutoff threshold is subjective, yet the majority of the primary studies set a cutoff of ≥5% of tumour cells (20 of the 26 studies) [22]. In one of these SRMA, subgroup analysis was conducted to standardize the cutoff value. Survival parameters were categorized by the cutoff percentage of PD-L1 positive cells (1%, 5%, 10%). OS and DSS did not show any statistically significant results when analysed by different cutoff points [22].

## 4. Discussion

The PD-L1 (also known as CD274 or B7H1) immune checkpoint has been extensively studied in HNSCC because it plays a critical role in cancer evading immune recognition [22]. Immunotherapy targeting the PD-L1/ PD-1 axis can be achieved with the use of monoclonal antibodies directed against either PD-1 (e.g., pembrolizumab, nivolumab) or PD-L1 (e.g., atezolizumab, avelumab, durvalumab). Among these, nivolumab and pembrolizumab have been approved for OSCC treatment [17]. However, CPIs can be associated with immune-related adverse effects (irAEs), such as pneumonitis, as described in one study [32]. Additionally, not all HNSCC patients respond favourably to immunotherapy [33]. Moreover, resistance to immunotherapy is another challenge that has been addressed and studied in HNSCC in recent years [34]. Effectiveness of this therapy depends on the careful selection of HNSCC cases based on successful biomarker assessment.

We did not find any systematic review specifically focused on PD-L1/PD-1 expression in OPMDs, highlighting the need for further investigation through a dedicated SR studies. One existing SRMA, which did not meet our inclusion criteria, examined PD-L1 expression alongside 172 other protein markers in the malignant progression of oral leukoplakia (OL) to OSCC. Although the study identified PD-L1, Mdm2, and Mucin-4 as significantly more abundant in OSCC, it did not establish PD-L1 as a standalone biomarker for risk prediction in OL. However, given the biological relevance of immune checkpoints in carcinogenesis, further systematic evaluation of their role in OPMDs remains warranted. [35]. Another systematic review study assessed the PD-L1/PD-1 axis in OPMDs but also evaluated it alongside other biomarkers. Additionally, it did not involve a meta-analysis, which contradicts our inclusion criteria. This study demonstrated that PD-1 and PD-L1 were expressed in most OPMDs and OSCC samples, with their presence being associated with higher disease progression and lower survival rates. However, the study concluded that limited data are available on the relationship between PD-L1/PD-1 and OPMDs [18].

Dysplastic lesions with PD-L1 expression on epithelial and subepithelial cells can escape the host immune response. Therefore, Blocking the PD-1/PD-L1 axis may help prevent the malignant transformation of OPMDs [21,36]. One clinical study demonstrated a relationship between the PD-L1/PD-1 axis and OPMDs, showing elevated PD-L1 levels in leukoplakia and oral submucous fibrosis (OSMF) cases compared to healthy individuals [37]. Another study found that PD-L1 levels increased in dysplastic epithelial cells as premalignant lesions progressed to malignancy [38].

Individualized immunotherapy for OSCC patients depends on identifying biomarkers that are upregulated in oral cancer cases. Twelve immune checkpoints have been studied for their relationship with OSCC. Seven of them, including FKBP51, B7-H4, B7-H6, ALHD1, PD-L1, B7-H3, and IDO1, have been associated with worse survival among OSCC patients in at least one study. Five molecules (CTLA-4, TLT-2, VISTA, PD-L2, and PD-1) did not show significant prognostic value [39].

Understanding the role of the PD-L1/PD-1 axis in oral squamous cell carcinoma (OSCC) is crucial for exploring personalized immunotherapeutic treatments. For instance, in one study involving 120 patients pathologically diagnosed with OSCC, half received anti-PD-1 therapy. This treatment enhanced T-cell immunity, improved patient survival, and promoted tumour regression [40].

Levels of PD-L1/PD-1 expression in relation to overall survival could potentially inform personalized treatment decisions and serve as a valuable biomarker in those patients.

However, the prognostic role of PD-L1 in OSCC cases has shown controversial results. Some studies reported worse overall survival of OSCC patients with high levels of PD-L1, while others reported a lack of prognostic value for this biomarker [22,30]. These controversial results attributed to heterogeneity between studies, which can be connected to many factors related to IHC. There are different IHC techniques utilized to detect PD-L1 and PD-1 expression, including the antibody used, staining technique, type of assay, and cutoff value. There are two common scores utilized for detecting a positive PD-L1 biomarker, which are tumour proportion score (TPS) and combined positive score (CPS). The TPS refers to the proportion (%) of PD-L1 positive tumour cells in correlation to the total number of viable tumour cells × 100; the combined positive score (CPS) refers to the number of all PD-L1-positive cells (both neoplastic and inflammatory cells) relative to the number of all viable tumour cells × 100 [41]. A CPS of less than 1 is considered as negative, while CPS ≥ 1 and ≥20 are regarded as positive [42]. Moreover, positive expression can be further scored into low (≥1–49) and high expression (≥50) [43]. Tumours with TPS scores of 50% or higher tend to respond better to immune checkpoint inhibitor (ICI) treatment alone, while those with lower TPS percentages might benefit more from a combination of ICI treatment and chemotherapy [44].

CPS is a better predicator of overall survival than TPS in patients with advanced non-small cell lung cancer (NSCLC) being treated with immune check point monotherapy. Notably, the improved performance was attributed to the TPS−/CPS+ subgroup, suggesting that CPS might be a more effective in predicting biomarkers for assessing ICI efficacy [41]. Moreover, different assays were utilized.

Four FDA approved PD-L1 immunohistochemical assays use four distinct PD-L1 antibodies, including 22C3, 28–8, SP263, and SP142, on two IHC platforms (Dako and Ventana), each with its own scoring system [45]. Particularly, the 22C3 PharmDx assay and SP263 assay have been utilized in HNSCC cases [46]. According to one study, results from these assays at cutoffs ≥1 and ≥20 demonstrated high agreement in head and neck cancer [46]. However, another study indicated that when categorizing patients based on clinically relevant cutoffs, notable differences between the assays were evident: agreement was lacking for both TPS and CPS. The SP263 assay stained a greater proportion of cells compared to other assays, especially when using the CPS [23].

Antibodies also contribute to inter-study variability, as demonstrated in one of the included SRMAs [22]. This analysis showed that out of all the antibodies, only two antibodies (E1L3N and 5H1) were employed in more than one study measuring the same outcome. The 5H1 antibody was used in two studies assessing overall survival and was associated with a worse prognosis (HR = 2.63; 95% CI = 1.15–6.02; *p* = 0.02). The E1L3N antibody was utilized in four studies of disease-specific survival (DSS), including 543 patients, and showed a statistically significant result in terms of worse prognosis (HR = 2.19; 95% CI = 1.41–3.39; *p* = 0.0004). The methodologies considered were membranous-only staining (10 studies), cytoplasmic or membranous staining (7 studies), and cytoplasmic or nuclear staining (2 studies) [22].

Manual scoring of PDL-1 IHC slides is another factor contributing to variability, as pathologists’ interpretations can introduce a potential error. The growing integration of digital pathology and artificial intelligence into routine laboratory assessment presents a chance for standardization and to utilize technologies for enhancing the clinical utility of PD-L1 IHC assays [45]. Digital pathology technologies are being investigated, such as random forest/supervised learning [24], supervised learning [47], and GAN/semi-supervised learning [48].

A second factor includes unresolved issues about how to detect positive PD-L1 expression, whether on tumour cells or inflammatory cells [22]. Interestingly, a systematic review and meta-analysis of 17 studies involving 3190 patients with HNSCC found that a high level of PD-L1 expression exclusively on immune cells were associated with improved survival in patients with localized and locoregionally advanced HNSCC, while its expression on tumour cells was not correlated with survival [49].

HPV infection is a risk factor of HNSCC. Patients with HPV-positive tumours have better prognosis than negative ones. Moreover, HPV-positive patients showed better responses for immunotherapy, including pembrolizumab [6,50]. In addition, those patients with HPV positivity and PD-L1 expression showed a better overall survival outcome [51]. The proposed effectiveness of immunotherapy in HPV-positive HNSCC is mainly associated with the characteristics of HPV-positive tumours, including immune cell infiltration, tumour immunogenicity, and expression of immune checks [52]. In our review, only two SRMAs [17,31] investigated the relationship between PD-L1 and HPV status in OSCC cases, and they found significant and high level of expression of PD-L1 in OSCC HPV-positive cases. There is a significant relationship between high level of PD-L1 in female patients in four out of the five SRMAs. The reason behind this phenomenon remains unclear, but it has been noted that the expression of PD-L1 can be upregulated by estrogen [22]. Estrogen, a hormone predominantly found at higher levels in women, is known to influence various immune responses, potentially enhancing immune evasion mechanisms in tumours. Estrogen has been shown to directly interact with estrogen receptors (ERs) on immune cells, such as T-cells and macrophages, as well as on tumour cells. This interaction could promote the upregulation of PD-L1 expression on the surface of tumour cells, contributing to the immune evasion mechanism and potentially influencing the response to immunotherapy. One study demonstrated that estrogen receptors could enhance PD-L1 expression in breast cancer cells, thereby promoting immune suppression in a hormone-dependent manner [53].

While high PD-L1 expression has been correlated with lymph node metastasis [22], its role as a prognostic marker remains complex. While PD-L1 may influence disease progression, its prognostic significance is confounded by the dynamic interaction between tumour cells and the host immune system.

A limitation of this study is that our conclusions are based on the results of primary studies involved in these SRMAs, which exhibited high heterogeneity. Moreover, overlap between primary studies in some SRMAs can inflate sample sizes and event counts, making the effect appear more precise. Additionally, SRMAs authors sometimes present their findings without properly discussing the results.

## 5. Conclusions

Our study revealed a lack of evidence for the prognostic value of PD-L1 in improving overall survival in OSCC patients; but, its high expression is associated with specific clinicopathological features, including sex, lymph node metastasis, and HPV status. This highlights the potential for further exploration of personalized treatment strategies targeting the PD-L1/PD-1 axis, particularly in relation to these features, to improve therapeutic outcomes in those patients. These results are inconclusive due to the high heterogeneity of all included studies. We suggest addressing this issue by conducting new standardized SRMAs that include only high-quality studies with low risk of bias. These studies should standardize IHC techniques to ensure reliable and reproducible results. There should be standardization in the cutoff value used, type of antibody used, method of staining, and adoption of digital pathology and artificial intelligence to minimize personal errors in histopathological diagnosis. We were unable to find any systematic review and meta-analysis (SRMA) studies that solely focus on the role of the PD-L1/PD-1 axis in OPMDs. Therefore, there is an urgent need for well-planned systematic reviews on this subject.

## Figures and Tables

**Figure 1 jpm-15-00126-f001:**
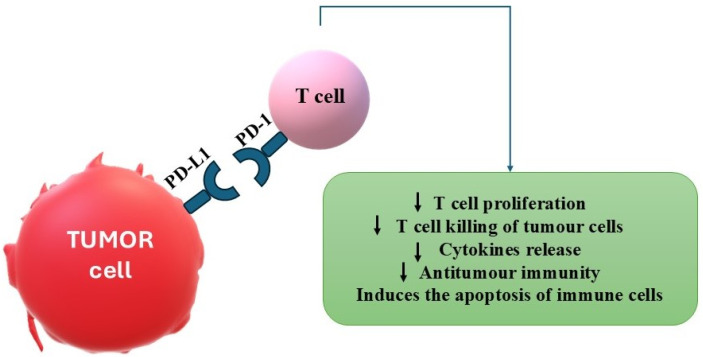
Diagram illustrating the role of PD-L1 in antitumor immunity (original illustration by the authors).

**Figure 2 jpm-15-00126-f002:**
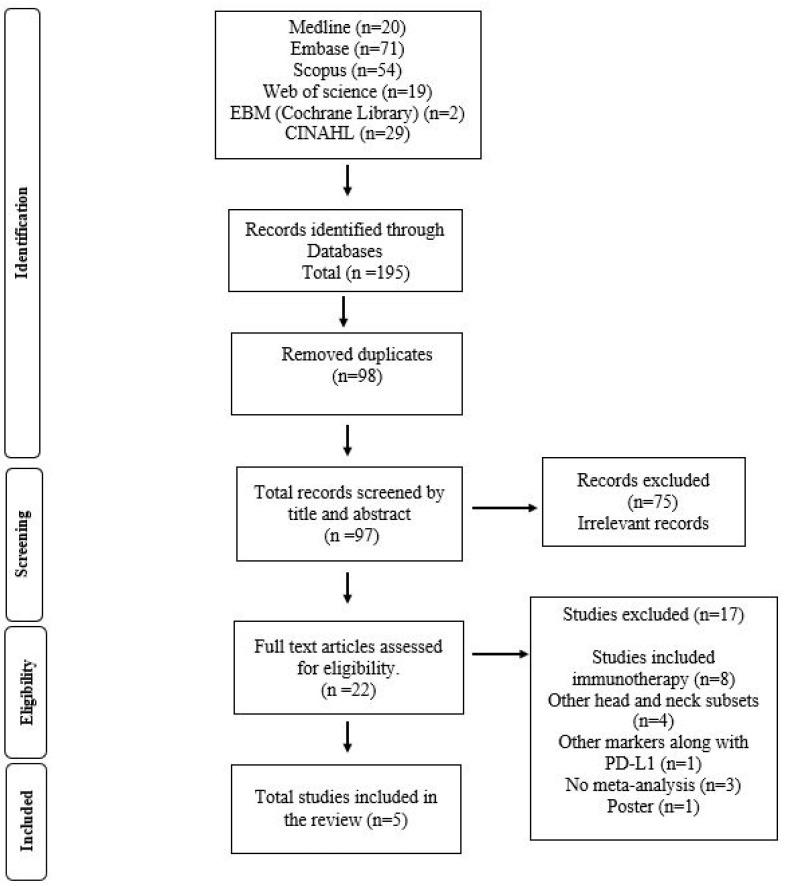
Flow chart showing the process of the search involved in this umbrella review for PD-L1 expression in OPMDs and OSCC.

**Figure 3 jpm-15-00126-f003:**
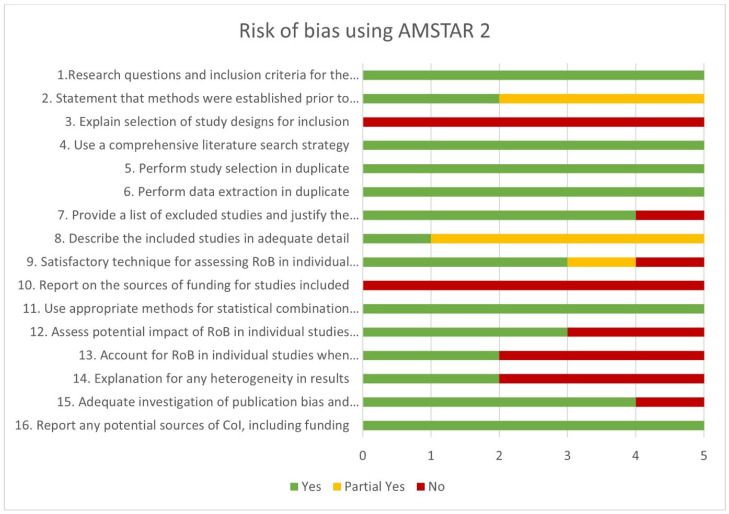
Risk of bias for the included systematic reviews and meta-analysis studies (SRMAs) utilizing the AMSTAR 2 tool. Number of SRMA studies (*n* = 5).

**Table 1 jpm-15-00126-t001:** Factors in immunohistochemistry and scoring methods contributing to PD-L1 expression heterogeneity in oral squamous cell carcinoma (OSCC) studies.

Causes of Variability in PD-L1/PD-1 Expression in Tissue Samples of Oral Squamous Cell Carcinoma (OSCC)	
Factor	Description
Study sample	Population diversity
IHC Techniques	Various IHC techniques including differences in the antibody used, staining technique, and assay type can lead to inconsistent findings.
Cutoff Value	Different studies use varying cutoff values to classify PD-L1 expression as positive or negative, which affects the interpretation of results.
Scoring technique	Different scoring methods impact outcome reporting. Tumour Proportion Score (TPS) measures PD-L1 positivity in tumour cells, while Combined Positive Score (CPS), which includes both neoplastic and inflammatory cells, is more commonly used and linked to better prognostic outcomes.
Human factors	Pathologists’ interpretations can introduce a potential error

**Table 2 jpm-15-00126-t002:** Characteristics of the SRMAs included in this umbrella review, which assessed PD-L1 expression in OSCC tissue. n: number, HR: hazard ratio (HR), CI: confidence interval, OS: overall survival, and NS: not significant. When there was significant heterogeneity in the studies, the pooled analysis was estimated using a random-effects model; when no apparent heterogeneity was found in the studies, a fixed-effect model was applied. High levels of heterogeneity were present in the assessed studies. Heterogeneity between studies was identified according to the criteria of *p* < 0.10 or I^2^ > 50%.

First Author Surname	Study (n)	Sample Size	Type of Study	Metanalysis Method Used	*p* Value, CI, and HR for Overall Survival (OS)	*p* Value, CI, and HR for Disease-Free Survival (DFS)	*p* Value, CI, and HR for Disease-Specific Survival (DSS)	*p* Value, CI, and HR for Lymph Node Metastasis	*p* Value, CI, and HR for PD-L1 and Sex	**HPV Status**	**Main Findings**
Troiano et al., 2019 [30]	10	1060	Prospective and retrospective clinical cohort	Random- effects model	7 studies assessed (HR, 0.60; 95% CI = 0.33–1.10; *p* = 0.10) NS I^2^ = 89%	(HR, 0.62; 95% CI = 0.21–1.88; *p* = 0.40) NS	(HR, 2.05; 95% CI = 0.53–7.86; *p* = 0.29) NS	(HR, 1.15; 95% CI = 0.74–1.81; *p* = 0.53) NS	High expression of PD-L1 is two times more frequent in female patients (OR, 0.5; 95% CI = 0.36–0.69; *p* < 0.0001) I^2^ = 0%	N/R	High PD-L1 expression did not correlate with poor prognosis of OSCC patients
He et al., 2020 [31]	23	3217	N/R	Random- effects model used for OS, sex, and HPV Fixed-effect model used for lymph node metastasis	17 studies assessed, 2435 patients (HR = 1.00; 95% CI = 0.76–1.30; *p* = 0.284) NS	N/R	N/R	Significant correlation (RR = 0.83; 95% CI = 0.76–0.91; *p* < 0.001) No apparent heterogeneity	18 studies assessed, positive PD-L1 expression was significantly higher in females (RR = 1.22; 95% CI = 1.07–1.38; *p* = 0.002) I^2^ = 56%	8 studies included, high PD-L1 was significantly correlated with HPV status (RR = 1.30; 95% CI = 1.04–1.62; *p* = 0.019) I^2^ = 69.3%	High PD-L1 expression was not related to OS. However, it was significantly related to sex, histological differentiation, TNM stage, and HPV infection status
Lenouvel et al., 2020 [22]	26	2532	N/R	Random- effects model	13 studies assessed, 1380 patients (HR = 1.00; 95% CI = 0.75–1.35; *p* = 0.98). NS I^2^ = 74% (*p* < 0.00001)	5 studies assessed, (HR = 1.42; 95% CI = 0.88–2.28; *p* = 0.15) NS I^2^ = 56% (*p* = 0.05)	8 studies, a statistically significant result was achieved (HR = 1.54; 95% CI = 1.03–2.28; *p* = 0.03) I^2^ = 58% (*p* = 0.01)	15 studies, 1707 patients, close to significant (OR = 1.35; 95% CI = 0.97–1.88; *p* = 0.07) I^2^ = 47% (*p* = 0.02)	4 studies, 1683 patients, PD-L1 overexpression was more likely in females (OR = 0.69; 95% CI = 0.53–0.91; *p* = 0.008)	N/R	PD-L1 expression was not related to OS. Worse prognosis with high PD-L1 in the cell membrane with a cutoff ≥5%, as measured by DSS and DFS
Yong-Xin Cui1 and Xian-Shuang Su, 2020 [17]	16	1989	Retrospective	Fixed-effect model	N/R	N/R	N/R	13 studies, significant correlation (N1–N3: RR = 1.19; 95% CI = 1.06–1.33; *p* = 0.003). I^2^ = 40.6% (*p* = 0.063)	15 studies, 1947 patients, high PD-L1 expression and female sex (RR = 1.28; 95% CI: 1.16–1.42; *p* < 0.001). I^2^ = 23.0% (*p* = 0.199)	8 studies, high PD-L1 expression and HPV-associated OSCC (RR = 1.38; 95% CI: 1.14–1.68; *p* = 0.001). I^2^ = 59.6%	High PD-L1 expression was correlated with clinicopathological features
Nocini R et al., 2022 [11]	12	1166	Prospective and retrospective	N/R	6 studies, 649 patients, lack of prognostic role of PD-L1 (HR for OS = 0.97; 95% CI = 0.53–1.80) NS	HR for DFS = 0.83; 95% CI = 0.47–1.46	N/R	N/R	N/R	N/R	A lack of prognostic significance of PD-L1 in OSCC

## Data Availability

The original data presented in this study were compiled from publicly available review articles and are openly accessible.

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
