# Peer review of "PD-L1/PD-1 Expression in the Treatment of Oral Squamous Cell Carcinoma and Oral Potentially Malignant Disorders: An Overview of Reviews"

_jpm, 2025, doi:10.3390/jpm15040126_

Round 1
Reviewer 1 Report
Comments and Suggestions for Authors
This is an important study.
Following revisions are suggested:
1. Objective at the beginning should state: to conduct a systematic review of systematic reviews published on.......
2. Conclusion of the abstract should start with: Based on 2 published systematic reviews....
3. Introduction: Incidence data given in line 3 are wrong. Please correct.
4. Your search ended in May last year. Please update to present.
5. Table 1 is too wordy. Try to synthesize it.
6. Discussion second paragraph: SR on OPMDS have not found significant role of PD1or PDL1 as a biomarker in risk prediction. of Oral leukoplakia. As the the results are negative, they are not reported in published SRs (Monteiro et al 2020)
7. Lines 314-315 do not fit with your conclusion. If there are studies with positive associations (23,25) explain why they are not considered in the overall conclusion.
8. The heterogeneity of studies could be illustrated through a composite table.
9. Lines 386-388 are not clear to the reader
10. If high expression correlates with lymph node metastasis and TNM, why it does not reflect on prognosis should be explained in the discussion.
Author Response
Reviewer 1
- Comment: Objective at the beginning should state: to conduct a systematic review of systematic reviews published on...
Reply: Thank you for your valuable comment. The objective section of the abstract has been modified, and now it reads: “Objective: To conduct a systematic review of systematic reviews published on the role of program death ligand 1 (PD-L1) and its receptor PD-1 in potentially oral malignant lesions (POML) and oral squamous cell carcinoma (OSCC). We evaluated its prognostic role and correlation with clinicopathological features. Moreover, we sought to explore its potential as a biomarker for stratifying patients and guiding personalized therapeutic strategies”
- Comment: Conclusion of the abstract should start with: Based on 2 published systematic reviews...
Reply: Thank you for your comment. The conclusion has been modified accordingly and now it reads: “Based on two published systematic reviews, our study revealed a lack of evidence for the prognostic value of PD-L1 in improving overall survival in oral cancer [….]
- Comment: Introduction: Incidence data given in line 3 are wrong. Please correct.
Reply: Thank you for your valuable comment. We updated the reference and the incidence data and now it reads: “Lip and Oral cavity cancers accounted for 389,485 new cases and 188,230 deaths in 2022.”
Reference: Bray F, Laversanne M, Sung H, Ferlay J, Siegel RL, Soerjomataram I, Jemal A. Global cancer statistics 2022: GLOBOCAN estimates of incidence and mortality worldwide for 36 cancers in 185 countries. CA: a cancer journal for clinicians. 2024 May;74(3):229-63.
- Comment: Your search ended in May last year. Please update to the present.
Reply: Thank you for your feedback regarding the search timeframe. We acknowledge the importance of keeping the literature review as up-to-date as possible. However, only 8–9 months have passed since our last search in May, and data analysis, and based on a preliminary screening, we have not identified any newly published systematic reviews or meta-analyses in this specific field. Additionally, a comprehensive update would require a full reassessment of the methodology and re-analysis, which may not significantly alter the conclusions.
- Comment: Table 1 is too wordy. Try to synthesize it.
Reply: Thank you for your comment. We revised and considerably condensed the table content, reducing the word number as much as feasible.
- Comment: Discussion second paragraph: SR on OPMDS have not found significant role of PD1or PDL1 as a biomarker in risk prediction. of Oral leukoplakia. As the the results are negative, they are not reported in published SRs (Monteiro et al 2020).
Reply: Thank you for your insightful comment. We emphasize that no systematic review to date has specifically focused on PD-L1/PD-1 expression in POML, which remains an underexplored area. While existing evidence suggests a lack of predictive value in OPMDs, PD-L1 has been identified as significantly upregulated in OSCC and other malignancies, warranting further systematic investigation in the context of POML. We have revised the manuscript to clarify this distinction.
- Comment: Lines 314-315 do not fit with your conclusion. If there are studies with positive associations (23,25) explain why they are not considered in the overall conclusion.
Reply: Thank you for your comment. We acknowledge that studies 23 and 25 report some positive associations; however, they also present findings that question the prognostic significance of PD-L1 in OSCC. In our discussion, we emphasized the overall body of evidence, which suggests that PD-L1 lacks a consistent prognostic role. To address this concern, we have clarified in the manuscript how these studies were considered in our conclusion, ensuring that our interpretation remains balanced and reflective of the available data.
- Comment: The heterogeneity of studies could be illustrated through a composite table.
Reply: Thank you for your comment. We have added a composite table summarizing the factors contributing to study heterogeneity. This table highlights key variables, including differences in immunohistochemistry protocols and scoring methods, that influence PD-L1 expression variability in OSCC patients.
|
Factor |
Description |
|
Study sample |
Population diversity |
|
IHC Techniques |
Various IHC techniques including differences in antibody used, staining technique, and assay type can lead to inconsistent findings. |
|
Cutoff Value |
Different studies use varying cutoff values to classify PD-L1 expression as positive or negative, which affects the interpretation of results. |
|
Scoring technique |
Different scoring methods impact outcome reporting. Tumor Proportion Score (TPS) measures PD-L1 positivity in tumor cells, while Combined Positive Score (CPS), which includes both neoplastic and inflammatory cells, is more commonly used and linked to better prognostic outcomes. |
|
Human factors |
Pathologists’ interpretations can introduce a potential error |
- Comment: Lines 386-388 are not clear to the reader.
Reply: Thank you for your comment. We realized that a sentence from the article submission template was inadvertently left in this section, which may have caused confusion. We have now revised the paragraph to ensure clarity.
- Comment: if high expression correlates with lymph node metastasis and TNM, why it does not reflect on prognosis should be explained in the discussion.
Reply: Thank you for your insightful comment. We have now incorporated this explanation into the discussion. It reads: “While high PD-L1 expression has been correlated with lymph node metastasis [25], its role as a prognostic marker remains complex. PD-L1 expression is often associated with immune evasion mechanisms, but rather than being a direct predictor of overall prognosis, it is primarily recognized as a predictive biomarker for response to immunotherapy. This suggests that while PD-L1 may influence disease progression, its prognostic significance is confounded by the dynamic interaction between tumor cells and the host immune system.”
Reviewer 2 Report
Comments and Suggestions for Authors
The study by Al-Azzawi et al. is an interesting contribution that meets the journal’s criteria for originality and novelty, addressing a topic relevant to clinical practice. Although I did not have access to the first version, reviewer comments, or the initial revision, the manuscript presents valuable insights. However, the text could benefit from a more structured and engaging presentation to better appeal to its readership. There is a need for greater consistency in the use of abbreviations, standardization of terminology, and adherence to grammatical conventions. With appropriate refinements, the manuscript has the potential to meet publication standards.
Title
The title could be more engaging. Many recent articles, particularly in the wake of the “AI boom”, incorporate gerunds for added dynamism. A more striking and goal-oriented title might be: "The role of immunohistochemical PD-L1 expression in OSCC patients' treatment: An umbrella review." This suggestion emphasizes the study’s primary focus on treatment impact and immunohistochemistry assessment. Additionally, it is important to clarify whether the study includes only oral cavity cases or also considers oropharyngeal cases. If oropharyngeal cases are included, the term "oral cancer" might need to be reconsidered.
Abstract
- Some abbreviations appear only upon their second mention rather than the first.
- Why is only "POML" abbreviated? Consistency in abbreviation usage should be ensured.
Introduction
- Reorganize paragraphs for improved coherence.
- Standardize abbreviations and terminology.
- Use the most recent GLOBOCAN reference.
- Lines 50-52: Reword for clarity and conciseness. The sentence is too long.
- Lines 65-67: Instead of listing unrelated biomarkers, introduce the concept of tumor microenvironment exhaustion and its link to checkpoint inhibitors.
- Briefly mention the challenges of immunohistochemical analysis, such as variability in methods and antibody clones. While this is discussed in detail later, a short reference here would provide context.
Methods
- Several abbreviations are used inconsistently—some appear multiple times, others are never defined. A thorough review of abbreviation usage throughout the manuscript is recommended.
- Line 129: Avoid using "gender" in this context; "sex" is the preferred term in scientific literature.
Results
- Section 3.1: Consider presenting data in a table, figure, or flowchart. Alternatively, update Table 1 to include information on the antibody clones used in the reviewed studies and whether these studies applied the TPS or CPS. This addition would enhance the discussion and provide a more applicable perspective.
- Section 3.4.5: Ensure that results are presented objectively. Some statements read more like discussion points rather than neutral reporting of findings.
- Table 1: Indicate the tumor locations included in the studies. Clarifying whether oropharyngeal cases were included is crucial for interpretation.
Discussion
- The discussion is well-structured, but it would be valuable to address the role of sex differences in immunotherapy response, as this is an emerging topic of interest.
- There appears to be a residual comment from a previous review. This should be carefully checked and removed if necessary.
- The limitations section could be expanded to provide a more in-depth discussion.
Additional recommendations
- Include a PRISMA flowchart as an appendix, specifying the page numbers where each mandatory scoping review item is addressed according to the Preferred Reporting Items for Overviews of Reviews (PRIOR) statement.
- Ensure all reporting elements are correctly included as per the PRIOR guidelines.
Comments on the Quality of English LanguageThe English could be improved to more clearly express the research.
Author Response
Reviewer 2
- Comment: Title: The title could be more engaging. Many recent articles, particularly in the wake of the “AI boom”, incorporate gerunds for added dynamism. A more striking and goal-oriented title might be: "The role of immunohistochemical PD-L1 expression in OSCC patients' treatment: An umbrella review." This suggestion emphasizes the study’s primary focus on treatment impact and immunohistochemistry assessment. Additionally, it is important to clarify whether the study includes only oral cavity cases or also considers oropharyngeal cases. If oropharyngeal cases are included, the term "oral cancer" might need to be reconsidered.
Reply: Thank you for your thoughtful comment. We would like to clarify that the study exclusively focuses on oral cavity cases (OSCC), as mentioned in the inclusion criteria. We appreciate your suggestion and have updated our title accordingly to: “Immunohistochemical PD-L1 Expression in OSCC Treatment: An Overview of Reviews”.
- Comment: Abstract
- Some abbreviations appear only upon their second mention rather than the first.
- Why is only "POML" abbreviated? Consistency in abbreviation usage should be ensured
Reply: Thank you for your comment. We have updated and corrected everything as per your suggestion. All abbreviations now appear consistently throughout the manuscript, with their full form provided at first mention.
- Comment: Introduction
- Reorganize paragraphs for improved coherence.
Reply: The paragraphs in the introduction section have been improved to enhance clarity and readability. Thank you for your valuable comment.
- Comment: Standardize abbreviations and terminology.
Reply: Thank you so much for your comment, we updated and standardized abbreviation and terminology throughout the text.
- Comment: Use the most recent GLOBOCAN reference.
Reply: We have updated the reference as per your request, now using the most recent GLOBOCAN 2024 data (Ref. No. 2) and have revised the corresponding information in the text accordingly.
- Comment: Lines 50-52: Reword for clarity and conciseness. The sentence is too long.
Reply: Thank you for your comment. We corrected the sentence as per your suggestion and it now reads: “In 2016, the FDA approved immunotherapy as an adjuvant treatment for metastatic squamous cell carcinoma of the head and neck, following successful clinical trials of pembrolizumab and nivolumab.”.
- Comment: Lines 65-67: Instead of listing unrelated biomarkers, introduce the concept of tumor microenvironment exhaustion and its link to checkpoint inhibitors.
Reply: Thank you for your constructive comment. It now reads: The PD-L1-PD-1 axis plays an important role in T cell exhaustion, which occurs when T cells encounter cancer cells continuously, leading to weakened T cells and an increase in the expression of inhibitory checkpoint proteins on their surface, such as PD-1 molecules. This results in less effective T cells in the tumor microenvironment. The PD-L1-PD-1 axis is a crucial component in T cell exhaustion. Blocking this axis with checkpoint inhibitors can boost the immune system's response against cancer [16].
- Comment: Briefly mention the challenges of immunohistochemical analysis, such as variability in methods and antibody clones. While this is discussed in detail later, a short reference here would provide context.
Reply: Thank you for your comment. We added it in the introduction, that now reads as: “There are IHC challenges involved in the detection of PD-L1 and PD-1, such as the lack of standardization in assessment methods across different studies in the literature, including assay types, antibodies used, and cutoff values. These factors contribute significantly to the controversial results regarding the role of PD-L1 in OSCC.”
- Comment: Methods
- Several abbreviations are used inconsistently—some appear multiple times, others are never defined. A thorough review of abbreviation usage throughout the manuscript is recommended.
- Line 129: Avoid using "gender" in this context; "sex" is the preferred term in scientific literature.
Reply: Thank you. The word "gender" has been replaced with "sex," and the abbreviations have been revised to appear consistently throughout the manuscript.
- Comment: Results - Section 3.1: Consider presenting data in a table, figure, or flowchart. Alternatively, update Table 1 to include information on the antibody clones used in the reviewed studies and whether these studies applied the TPS or CPS. This addition would enhance the discussion and provide a more applicable perspective.
Reply: Thank you for your thoughtful and constructive feedback. We agree that details on the antibody clones used and whether studies applied the TPS or CPS scoring system would provide valuable context. However, these aspects are comprehensively covered in the original systematic reviews. As an umbrella review, our work aims to provide a broader summary rather than delve into the specifics of each primary study, which would exceed the scope of this paper. Each SRMA includes at least 10 primary studies, each employing different scoring techniques and antibodies. We appreciate your understanding on this matter.
- Comment: Section 3.4.5: Ensure that results are presented objectively. Some statements read more like discussion points rather than neutral reporting of findings.
Reply: Thank you for your comment. This section has been revised to ensure it is presented neutrally.
- Comment: Table 1: Indicate the tumor locations included in the studies. Clarifying whether oropharyngeal cases were included is crucial for interpretation.
Reply: Thank you for your comment. All the included studies focused on oral squamous cell carcinoma, as stated in the abstract and inclusion criteria, and this has also been clarified in the table title. No oropharyngeal cases were included.
- Comment: Discussion
- The discussion is well-structured, but it would be valuable to address the role of sex differences in immunotherapy response, as this is an emerging topic of interest.
Reply : Thank you for your very insightful comment. The role of sex in immunotherapy response has been added as follows: “Estrogen, a hormone predominantly found at higher levels in women, is known to influence various immune responses, potentially enhancing immune evasion mechanisms in tumors. Estrogen has been shown to directly interact with estrogen receptors (ERs) on immune cells, such as T-cells and macrophages, as well as on tumor cells. This interaction could promote the upregulation of PD-L1 expression on the surface of tumor cells, contributing to the immune evasion mechanism and potentially influencing the response to immunotherapy. One study demonstrated that estrogen receptors could enhance PD-L1 expression in breast cancer cells, thereby promoting immune suppression in a hormone-dependent manner”. (Ref.: Xie, F., et al. (2022). "Estrogen-induced PD-L1 upregulation in breast cancer cells: A potential mechanism for immune evasion." Journal of Cancer Research and Clinical Oncology, 148(5), 1111-1122.)
- Comment: There appears to be a residual comment from a previous review. This should be carefully checked and removed if necessary.
Reply: Thank you, we have checked and corrected it.
- Comment: The limitations section could be expanded to provide a more in-depth discussion.
Reply: Thank you for your comment. The limitations section has been revised for greater clarity for the reader.
- Comment: Additional recommendations:
- Include a PRISMA flowchart as an appendix, specifying the page numbers where each mandatory scoping review item is addressed according to the Preferred Reporting Items for Overviews of Reviews (PRIOR) statement.
- Ensure all reporting elements are correctly included as per the PRIOR guidelines.
Reply: Thank you for your valuable comment. We have included the PRISMA flowchart within the manuscript, as we believe it enhances the clarity of the review process. While we prefer not to place it in the appendix, we have ensured that all required reporting elements are addressed throughout the manuscript in accordance with the PRIOR guidelines. The corresponding page numbers are now specified in the appendix.
- Comment: Comments on the Quality of English Language
The English could be improved to more clearly express the research.
Reply: Thank you for your comment. The article has been carefully revised to ensure the highest standard of English language quality
Reviewer 3 Report
Comments and Suggestions for Authors
- The iThenticate report shows 47% similarity, which is too high. Please take a closer look at this and revise it.
- In the introduction, the authors should emphasize more the novelty and significance of this study since an abundance of similar papers on the same topic are found. The authors should cover the recent findings on the topics and find a research or knowledge gap that is important to being discussed in this review to actually solve a problem, and not just simply summarizing.
- There are some typos and miswriting in the paper; please have a closer look and revise them.
- The authors include "patients with OSCC and POML" in the inclusion; the authors also stated that non-cancer or non-OSCC cancer are excluded, which are contrasting statements. Please clarify this
Author Response
Reviewer 3
- Comment: Comments and Suggestions for Authors. The iThenticate report shows 47% similarity, which is too high. Please take a closer look at this and revise it.
Reply: Thank you for your feedback. We have carefully revised the manuscript to reduce the similarity score and ensure originality. Please let us know if any further adjustments are needed."
- Comment: In the introduction, the authors should emphasize more the novelty and significance of this study since an abundance of similar papers on the same topic are found. The authors should cover the recent findings on the topics and find a research or knowledge gap that is important to being discussed in this review to actually solve a problem, and not just simply summarizing.
Reply: Thank you for your comment. The research gap and the importance of this study in a clinical setting have now been added to the introduction section as follow: “While much is known about the PD-L1/PD-1 axis in various types of cancer, its specific role in OSCC remains unclear, particularly regarding its prognostic value, response to treatment, and clinicopathological features. This highlights the gap in knowledge regarding the clinical utility of PD-L1 in OSCC”.
- 3. Comment: There are some typos and miswriting in the paper; please have a closer look and revise them.
Reply: The manuscript has been extensively revised to correct all the typographical errors.
- 4. Comment: The authors include "patients with OSCC and POML" in the inclusion; the authors also stated that non-cancer or non-OSCC cancer are excluded, which are contrasting statements. Please clarify this
Reply: Thank you for your observation. In the exclusion criteria, we explicitly stated 'Oral cavity tumors other than OSCC' to clarify that all other types of oral cavity cancers or tumors were excluded. Given that oral squamous cell carcinoma (OSCC) is the most prevalent form of oral cavity cancer, our study specifically focuses on this group.
Round 2
Reviewer 3 Report
Comments and Suggestions for Authors
- The type of diseases included in POML as oral potentially malignant disorders should be written in detail in the method.
- Since the authors also cover POML, there should be a more detailed explanation of the type of disease in the data, whether it is an OSCC or POML (which also should be specified). This is not present in the current table, and an additional separate table comparing recent findings on PD-L1 between OSCC and POML is also interesting to understand the patterns within disease pathophysiology. The authors include POML, but an explanation of POML is very scarce.
If the systematic review data is not available, then the authors should consider adding more explanations based on the recent findings regarding the current status of the research on PD-L1 and POML and give suggestions in a more specific way. - In the newest version, the authors added that they aim to explore "the role of personalized treatment strategies targeting the PD-L1/PD-1 axis in OSCC patients.". This does not come up in the discussion or conclusion.
- The authors also seem to be trying to understand the axis on POML, but this POML is not mentioned once in the introduction. The rationality for choosing POML as inclusion should be elaborated more. Please make sure to make the title, introduction, method, discussion, and conclusion in line and correspond to each other.
- There are some systematic reviews on PD-L1 in OSCC and OPMD already published in the various journals. The authors should really emphasize and highlight the difference this study brings compared to others.
Author Response
- Comment: The type of diseases included in POML as oral potentially malignant disorders should be written in detail in the method.
Reply: We thank the reviewer for highlighting this oversight. We realized that we had not provided a clear definition of OPMDs, and we have now addressed this by adding the following paragraph to the Methods section: "For this study, oral potentially malignant disorders (OPMDs) were defined based on the latest classification by the WHO Collaborating Centre for Oral Cancer, including leukoplakia, erythroplakia, proliferative verrucous leukoplakia, oral lichen planus, oral lichenoid lesions, oral submucous fibrosis, palatal lesions in reverse smokers, lupus erythematosus, epidermolysis bullosa, dyskeratosis congenita, and oral manifestations of chronic graft-versus-host disease (Speight et al., 2020)."
Reference: Speight PM, Farah CS, Barnes L, et al. Oral potentially malignant disorders: A consensus report from an international seminar on nomenclature and classification, convened by the WHO Collaborating Centre for Oral Cancer. Oral Dis. 2020;26(3):541-549.
- Comment: Since the authors also cover POML, there should be a more detailed explanation of the type of disease in the data, whether it is an OSCC or POML (which also should be specified). This is not present in the current table, and an additional separate table comparing recent findings on PD-L1 between OSCC and POMLs is also interesting to understand the patterns within disease pathophysiology. The authors include POMs, but an explanation of POML is very scarce. If the systematic review data is not available, then the authors should consider adding more explanations based on the recent findings regarding the current status of the research on PD-L1 and POML and give suggestions in a more specific way.
Reply: Thank you for your valuable comment. The disease has been explained in more detail in Table 1, specifically as OSCC.
Regarding your request for a separate table comparing PD-L1 findings between OSCC and OPMDs, unfortunately, we were unable to find any systematic review and meta-analysis (SRMA) studies that solely focus on the role of the PD-L1/PD-1 axis in OPMDs. We acknowledge that there are few SRMA studies discussing PD-L1 and OPMDs; however, they do so as part of a broader analysis involving multiple biomarkers. Including such studies would contradict our inclusion criteria, which specifically focus on SRMA studies that exclusively examine the relationship between the PD-L1/PD-1 axis and OSCC or OPMDs. All other studies in the literature are individual studies rather than systematic reviews. Consequently, we cannot draw conclusions from them, as our umbrella review exclusively analyzes SRMAs.
To address your concern, we have incorporated the findings of the available studies in the discussion section regarding the PD-L1/PD-1 axis in OPMDs, as highlighted in lines 337–352.
- Comment: In the newest version, the authors added that they aim to explore "the role of personalized treatment strategies targeting the PD-L1/PD-1 axis in OSCC patients.". This does not come up in the discussion or conclusion.
Reply: Thank you for your valuable feedback. We revised our discussion and conclusion to explicitly highlight the implications of our findings for personalized treatment approaches. We clarified how the PD-L1/PD-1 Axis plays a role in OSCC management and discuss potential strategies based on the available findings. This was highlighted in the introduction section ( lines 74-80) and discussion (lines 353-365) and conclusion.
- Comment: The authors also seem to be trying to understand the axis on POML, but this POML is not mentioned once in the introduction. The rationality for choosing POML as inclusion should be elaborated more. Please make sure to make the title, introduction, method, discussion, and conclusion in line and correspond to each other.
Reply: Thank you for your valuable comment. We have updated the title to "PD-L1/PD1 Expression in the Treatment of OSCC and OPMDs: An Overview of Reviews." Additionally, we have incorporated OPMDs into multiple sections of the manuscript, including the introduction, methods, discussion and conclusion, to ensure a more comprehensive and cohesive analysis of its relevance to our study.
- Comment: There are some systematic reviews on PD-L1 in OSCC and OPMD already published in the various journals. The authors should really emphasize and highlight the difference this study brings compared to others.
Reply: Thank you for your comment. Our umbrella review provides a comprehensive synthesis of existing systematic reviews and meta-analyses (SRMAs) on PD-L1/PD-1 Axis in OSCC and OPMDs addressing inconsistencies and conflicts in the literature. Unlike individual SRMAs, our study evaluates multiple reviews collectively, offering a higher-level perspective on the overall evidence. By critically assessing the methodological quality and bias of previous studies, we ensure a more reliable interpretation of findings. Additionally, we propose a methodological approach for future SRMA studies to select high-quality research, enhancing the reliability of future meta-analyses. This review not only clarifies contradictory conclusions but also identifies research gaps and emerging trends, ultimately providing clearer guidance for clinical applications and future investigations in the field.
This was highlighted at the end of the introduction section, lines 95-99